# Vulnerability Assessment and Adaptation Strategies for the Impact of Climate Change on Agricultural Land in Southern Taiwan

**Kuo-Ching Huang \*** , **Chen-Jai Lee, Shih-Liang Chan and Cheng-Hsin Tai**

Department of Real Estate & Built Environment, National Taipei University, New Taipei City 237303, Taiwan; audio@gm.ntpu.edu.tw (C.-J.L.); slchan@mail.ntpu.edu.tw (S.-L.C.); shin@mail.ntpu.edu.tw (C.-H.T.)
\* Correspondence: kuoching.huang@gmail.com; Tel.: +886-2-8674-1111 (ext. 67436)

**Abstract:** Maintaining a certain amount of agricultural land and promoting its agricultural land utilization efficiency is essential in a country. Many innovative strategies for adapting to climate change have been implemented in developed countries. To achieve the goal of climate change adaptation for agricultural land, a vulnerability assessment of farmland is indispensable. Based on the research framework of the Intergovernmental Panel on Climate Change, this study applied the structure of exposure, sensitivity, and adaptation to build criteria and conduct an evaluation of a designated area in Southern Taiwan. We identified the key factors of the vulnerability of farmland, through mapping with spatial analysis, and by using geographic information system tools. The main purpose of the application of a vulnerability assessment is not to explicitly describe the status of agricultural land to climate change, but to help local government and farmers to identify the critical area, and to discuss the appropriated adaptive policies. According to the results of the vulnerability assessment of agricultural land, the entire study region can be divided into three patterns: Pattern 1, located in the western coastal zone, filled with various attributes of high vulnerability; Pattern 2, distributed on the central plain region in the east, with complete blocks of agricultural land and low vulnerability; and Pattern 3, located in the central plain region to the west, a region in which areas with various vulnerability levels. The following three types of adaptation strategies for climate change for farmland were established: (1) the enhancement of agricultural production, (2) the maintenance of agricultural production, and (3) the conservation of the agricultural environment. The current results can serve as valuable guidelines for governments to implement feasible local adaptation strategies in the future.

**Keywords:** agricultural land; vulnerability assessment; adaptation strategy; geographic information system; Taiwan

## 1. Introduction

Global climate change leads to the occurrence of various phenomena, such as an increase in temperature, change in rainfall frequency, rise in sea levels, decrease in snow-covered areas, and increase in the occurrence of extreme events. The Fourth Assessment Report (AR4) of the Intergovernmental Panel on Climate Change (IPCC) indicated that the impact of climate change can reduce global agricultural productivity, as well as regional water resource shortages in the future, particularly in island countries.

Taiwan is located in the subtropical island climate zone, and is therefore extremely sensitive to the effects of climate change. The resulting events may include damage to coastal areas from the rising sea levels, an increase in water supply-demand because of rainfall changes, reduction in agricultural

productivity, and the more frequent occurrence of natural disasters. Climate change critically affects various issues—one of the main issues being food security. To adjust to the impact of climate change on European agriculture, the European Union has proposed adjustments for crop production and farming systems, as well as proposed agricultural land use patterns exploiting agricultural land versatility, thereby balancing environmental, social, and economic functions in different European regions [1]. In this context, the approach for maintaining a certain quality and quantity of farmland resources to ensure domestic food self-sufficiency has become a strategic issue at the national level in Taiwan.

In general, responses to the impact of climate change have primarily focused on two dimensions: mitigation and adaptation. Mitigation strategies are concerned with actions resulting in decreased greenhouse gas emissions, whereas adaptation strategies focus on reducing vulnerability to climate change impact by using advanced mechanisms to predict and prevent the occurrence of disasters, thereby minimizing the effects of such disasters [2].

To promote the effective use of agricultural land resources in developed countries, an emphasis on agricultural development has been gradually more frequently considered in spatial planning processes and new strategy patterns. In these planning processes, the Adaptation Policy Frameworks (APFs) for Climate Change, put forward by the United Nations Development Program (UNDP) and Global Environment Facility in 2004, have been widely applied for practices conducted in various fields, and have become the main reference and relevant planning criteria specifications. The APFs not only help stakeholders understand the possible options for the implementation of adaptation programs, technology, and resources, but also facilitate the implementation of climate change adaptation programs by decision makers in local areas and various sectors and countries [3,4].

The APF implementation process comprises five components: (1) scoping and designing an adaptation project, (2) assessing current vulnerability, (3) assessing future climate risks, (4) formulating an adaptation strategy, and (5) continuing the adaptation process (Figure 1). Among these, assessing current vulnerability and assessing future climate risks evaluate current and future vulnerability trends and risks, indicating that vulnerability assessment is significant, with high research value for designing climate change adaptation strategies. Therefore, approaches for measuring the impact of climate change on agricultural land resources through vulnerability assessment are a crucial issue for adaptation strategies on agricultural land.

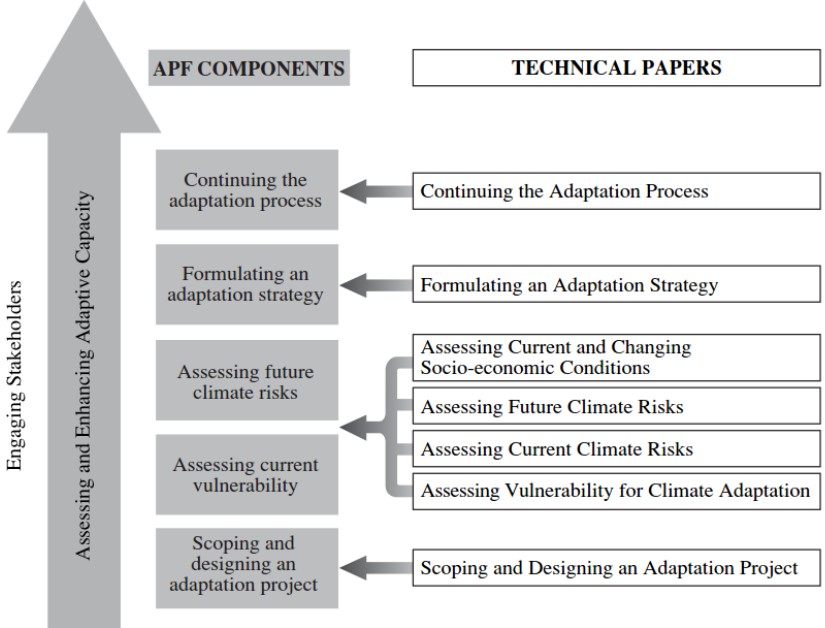

**Figure 1.** Adaptation Policy Framework (APF) Implementation Process and Basic Components. (United Nations Framework Convention on Climate Change, 2008, APF Technical Paper).

The vulnerability of agricultural land is mainly engendered by the climatic impact. The natural environment and climatic conditions are both strongly associated with agricultural productivity, directly affecting crop yield and quality [1,5]. Therefore, when external environmental changes caused by climatic factors affect the crop growth conditions of agriculture, agricultural production often decreases within inappropriate temperature and rainfall. This decrease also reduces the efficiency of agricultural land use, and affects the whole agricultural economy in advance, eventually resulting in a reduced number of farmers, loss in competitiveness, and food security problems, all in the agricultural sector.

Furthermore, climate change impact also affects the multiple functions of agricultural land in various ways (Table 1). Climate change causes serious environmental problems related to agricultural land, human activity, and land use. These effects are caused by not only extreme climatic events and disasters, but also by changes in climatic conditions. Thus, the agricultural land vulnerability can be defined as the climate change-induced decline in the effects of the multiple functions of agricultural land.

**Table 1.** Types of Climate Change Impacts Affecting Agricultural Land.

| Function | Impact Type | Description |
|---|---|---|
| Agricultural Production | Change in climatic conditions to grow crops | Climate change leads to changes in temperature and rainfall and directly affects crops. |
| | Increased demand for irrigation | Climate change alters rainfall patterns; erratic rainfall increases irrigation demand. |
| | Floods | Climate change causes extreme rainfall and increases flood frequency and strength, even external impacts as soil erosion, movement of nutrients and pesticides, salinization, water withdrawal, and groundwater contamination. |
| | Heatwaves and cold snaps | Changes in temperature and amplitude and frequency of heat waves or cold snaps result in yield loss. |
| Socioeconomic Factors | Changes in farming institutions | Agricultural land-use patterns are necessary for reallocating response to climate change impact. |
| | Implementation of agricultural facilities | Implementation of agricultural facilities increases the adaptation ability for climate change and the allocative efficiency of agricultural resources. |
| | Evolution of agroprocessing industry | To adapt to changes in climatic conditions, the agroprocessing industry attempts to reposition its functions through restructuring. |
| | Impact of agricultural culture | The sustainability of agricultural culture and the related agricultural industries requires adaptation polices to deal with the challenges of low competitiveness resulting from climate change. |
| Environmental Ecology | Impact on biodiversity | Changes in climate patterns affect the original conditions of natural environments and affect biodiversity. |

Source: Lee and Chan, 2012 [6].

Traditionally, vulnerability and risk assessment studies in the agricultural sector have mainly focused on agricultural production and technical departments [7–12], and have seldom considered agricultural land. However, the concept of the multiple functions of agricultural land has altered these traditional values. Agricultural land resources are often required for food production as a principle, but these are within socioeconomic and ecological functions to maintain the environment and local development [6,13].

This study adopted Southern Taiwan as its case study area and aimed to (1) integrate the concept of the multiple functions of agricultural land based on the research framework of vulnerability proposed by the IPCC [14], (2) develop a method for both assessing and mapping the vulnerability of agricultural land, (3) introduce the proposed adaptation strategies for the proper maintenance of agricultural land

resources, and (4) provide directions for governments to work with the local adaptive actions of climate change on farmland.

## 2. Case Study Area: Southern Taiwan

The case study area is located in southern Taiwan, including Yunlin county, Chiayi county, and Tainan city, etc., (Figure 2). Southern Taiwan, with a total area of approximately 5386 km$^2$, is the main food production area in Taiwan; its current agricultural area is approximately 308,981 ha, as shown in the green area in Figure 2. The cultivation areas for the critical crop of this area, paddy rice, covers approximately 63,325 ha, thereby occupying more than 20% of the total agricultural land in the study area (Table 2). Most agricultural areas are located in the tropical and subtropical climate zones. The topology mainly comprises plains, but some hills and mountains are located on the eastern side. Creeks, streams, and rivers are spread over the entire area, causing flooding problems during storms or heavy rain. Recently, those events have brought complex disasters and impacts on agricultural land and its surrounding areas, such as soil erosion in the upstream, seawater flooding, and land salinization in the western coast, drainage and water supply problems, and so on. It is necessary to examine the impact of climate change on agricultural development, for maintaining the ability of agricultural production and the requirement of food security in Taiwan.

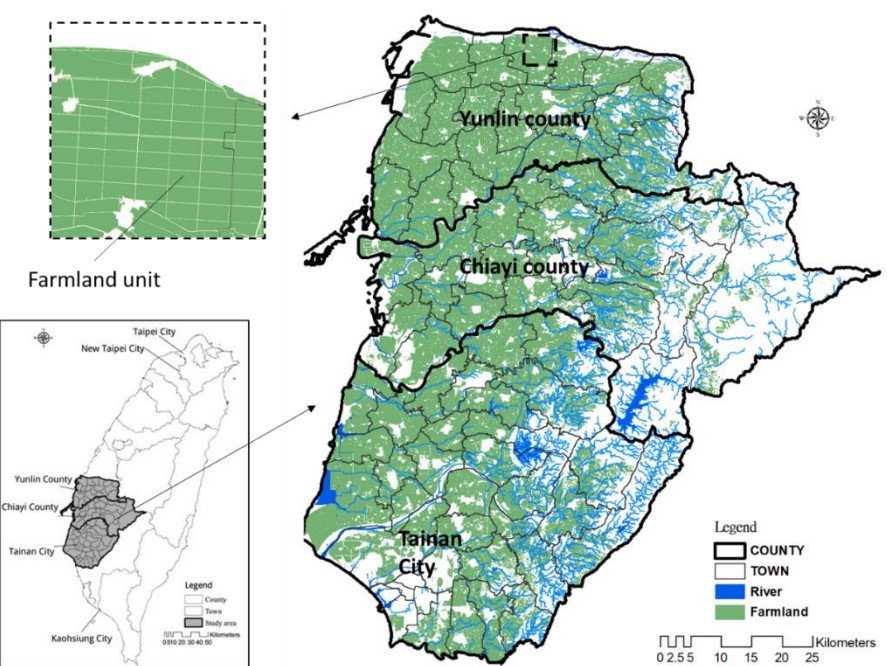

**Figure 2.** Location of the Study Area in Taiwan.

**Table 2.** Statistics of Main Crop Types in the Study Area (Unit: ha).

| Location | Crop | Area | Location | Crop | Area | Location | Crop | Area |
|---|---|---|---|---|---|---|---|---|
| Yunlin | Rice | 29,888 | Chiayi | Rice | 18,080 | Tainan | Rice | 15,355 |
|  | Sweet potato | 3209 |  | Corn | 4844 |  | Corn | 5414 |
|  | Potato | 1671 |  | Bamboo | 1914 |  | Sugarcane | 1286 |

## 3. Methodology and Materials

Vulnerability assessment, which typically discusses the environmental impact of natural disasters, has been widely used to investigate questions on natural disasters and climate change. The generally accepted definitions of vulnerability are those adopted by the United Nations Environment Programme (UNEP) and the IPCC [14,15]. The UNEP considers vulnerability to be the degree of loss caused

by potential damage, as well as a method for maintaining human welfare, when environmental, social, economic, and political systems are exposed to hazardous circumstances. The IPCC defines vulnerability as the degree of difficulty in maintaining stability within the system, when social and natural systems suffer climatic disasters. As a result of the different sensitivities of each system, the ability of each system to withstand climate change impact varies.

The main difference in the UNEP and IPCC definitions is based on the various exposed objects: The UNEP emphasizes human welfare, whereas the IPCC emphasizes the impact of exposure to disasters on social and natural systems. As such, the concept of vulnerability combines the potential impact and adaptation as its two major dimensions. Potential impact is derived from sensitivity and exposure to the whole environment. Adaptation behavior mitigates climate change impact related to disasters, and improves active adaptability to the environment by adjusting sensitivity and exposure intensities.

The aforementioned description indicates that agricultural vulnerability assessment must consider different aspects, different space scales, and different risk scenarios. In recent years, some considerable achievements have been noted in vulnerability assessment systems in the agricultural sector [7,8,10,11]. Gbetibouo and Ringler [10] proposed a vulnerability assessment framework and criteria for the agricultural sector.

The researchers divided agricultural vulnerability into potential impact and adaptation ability, as well as applying the aspects of physical capital—such as social aspects, human resources, financial sectors—regarding adaptation, and exposure and sensitivity factors regarding potential impact. Studies have focused on the analysis of different types of systems, including biological, socioeconomic, and agricultural systems [8]. These systems are based on the criteria of exposure, sensitivity, and adaptive criteria, as intended. Some of the aforementioned studies have focused on the application of system simulation methods, such as the groundwater loading effects of agricultural management systems. The authors investigated the effects of water and soil losses, water scarcity, and temperature changes on agricultural land. Moreover, the factors of vulnerability listed for climate change impact were mainly based on the characteristics of agricultural production; however, the suitability of agricultural land and the utilization of the aforementioned multiple functions were note considered. Agricultural land is one of the essential elements of agriculture. Emphasizing the multiple values of the nature of agricultural land (such as ecological diversity, educational value and location characteristics, etc.,) will help to enhance the possibility and application of policy inputs.

The Taiwan government has attempted to view the versatile features of agricultural land in terms of the concept of overall planning, and developed relevant adaptation strategies to cope with climate change impact. This study attempted to use simulation data obtained from different climate scenarios and explore the distribution of vulnerability on agricultural land in different contexts. The following sections describe the analysis framework, vulnerability indicator system, and data collection in this study.

### 3.1. Analysis Framework

This study followed the process of the simplified APFs, emphasizing the framework of vulnerability assessment, and then extended the applications for adaptation strategies. The simplified analysis framework is presented in Figure 3. The framework comprises three phases. Phase 1 reveals the study issues and objectives: here, we focused on food security and agricultural land management. Moreover, considering the aspects of ecology, environment, and landscape, maintaining the multifunctionality of agricultural land is extremely helpful for enhancing agricultural production, and is also relevant to the goal of sustainable agricultural development.

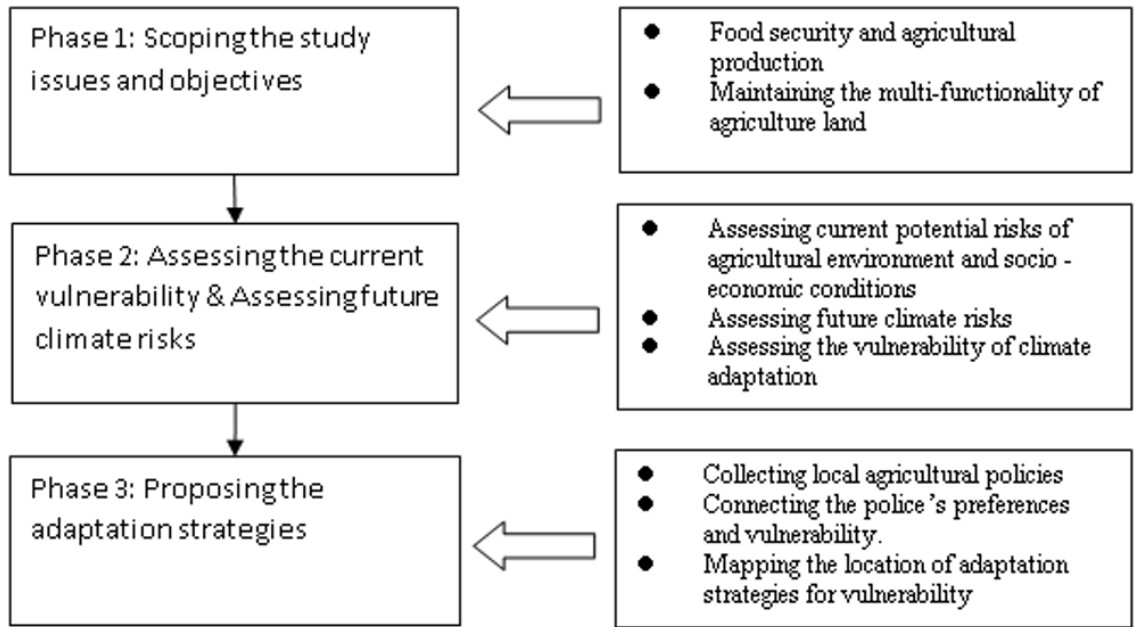

**Figure 3.** Analysis Framework of this Study.

Phase 2 focuses on the mechanism for assessing vulnerability and integrates the current level of vulnerability and future climatic risks through combination rules. Current potential risks are engendered by the physical environment, agricultural characteristics, and socioeconomic conditions. Determining future climatic risks, such as climate predictions and rising sea levels, proposed by the Taiwan government are based on the official simulation results [6,16].

In response to the current level of vulnerability and future climatic risks, Phase 3 involves the identification of adaptation policy options, as well as the formulation of these alternatives into a cohesive integrated strategy. The strategies for climate change adaptation in Southern Taiwan were proposed by integrating local agricultural policies and recognizing policy preferences. Finally, all of the proposed adaptation strategies were appropriately applied to areas with different vulnerability levels.

*3.2. Vulnerability Indicator System for Agricultural Land*

This study adapted the definition of vulnerability proposed by the IPCC (2007) as its basis. This definition includes exposure, sensitivity, and adaptation, all of which are based on the vulnerability assessment framework proposed by Rannow et al. [17]. The main components of the research framework include input data, classification, the creation of criteria, construction of an assessment matrix, and decisions regarding the level of vulnerability. The procedure was operated through various climate-based scenarios following this analysis process (Figure 4).

Overlapping and combination by rules are the two main approaches used for the entire analysis process. Overlapping aids in calculating the combinations of different scores through the indices of each dimension, such as sensitivity. A combination of rules can be applied to various consequences of different aspects or scenarios; for instance, it can be recombined into a result regarding the potential impact through exposure and sensitivity. Finally, the result of potential impact combines with adaptation by assessment matrix, to determine the distribution of agricultural land vulnerability. The vulnerability distribution will be used as a reference for constructing appropriate adaptation strategies and action plans.

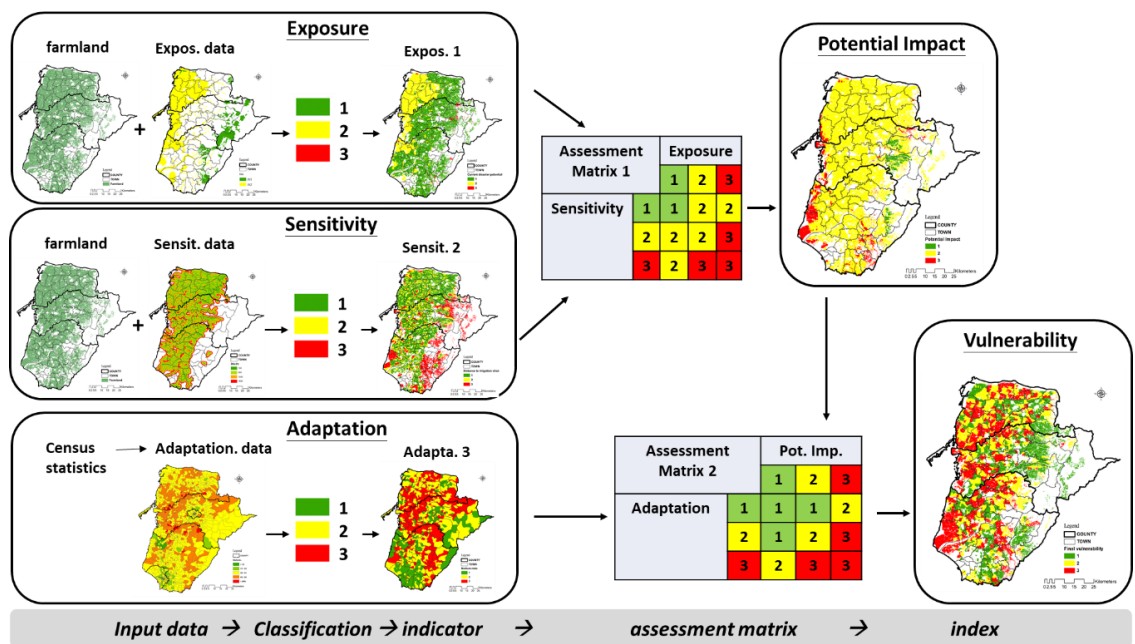

**Figure 4.** Vulnerability Assessment Structure for Agricultural Land. (Modified from Rannow et al. (2010)).

The types of vulnerability indices in the agricultural sector are quite diverse. This study attempted to define the agricultural vulnerability indicator system (Table 3) from the utilization of multifunctional farmland, especially in agricultural production, water resources, limitation of natural environment, and farmers' social-economic condition, etc. This system can be divided into three main aspects, which are further downscaled to a group level and finally to an item level. The criteria at the aspect level are exposure, sensitivity, and adaptation. The criteria at the group level are disaster potential, climate prediction, rising sea levels, soil, water, blocks, sensitive areas, household characteristics, agricultural organization, and socio-economic characteristics. Finally, the criteria at the item level are determined by the crucial attributes of each indicator group.

For computing the strength of the vulnerability, the score must be given according to its properties. Firstly, the farmland unit (see Figure 2) is used as an analysis unit, which integrated all input data involved indices. However, each indicator has specific measurements, causing difficulties in integration and computation. To resolve this problem, the sequential method is applied for measuring the score by using a relative level for each indicator, denoting that the score of each item represents the relative level of vulnerability. A higher score denotes a higher level of vulnerability. Moreover, to operate the measurement method consistently, this study used the grouping classification method "quantile" in ESRI ArcGIS software to divide the value of each item into three levels, and give each item a new score by its sequence. For instance, the score of 1 represents the lowest level of vulnerability, whereas a score of 3 represents the highest level. Regarding the alternative indicator, if it stands in a dangerous situation, the score is 3; the score is 1 if it does not stand in a dangerous situation.

To explicitly describe the calculation process, it explains the combination approach and composed levels between items, groups, and aspects. The weighting is equal. The weighting among criteria in the paper is different from other processes because the weighting value could not be significant, due to the many criteria to overlay. This article applied the rule-based approach suggested by an expert meeting for computing vulnerability. Based on this, the level composition approach is used for computing the score of items and group. For instance, the level group is composed of a level item, such as "Farmer household characteristics" group contains "Number of villager households/village area," "Age of person who commands cultivation", "Educational degree", and "Average agricultural income of villager household". The initial score of the group "Farmer household characteristics" is the

summary of scores from the four factors. Each farmland unit has its own summary score of the group. After reranking by the summary score and quantile analysis, the final score of the group "Farmer household characteristics' has been given by 1, 2, 3. The level composition approach is applied in the score of sensitivity aspect, soil group, water group, blocks group, sensitive areas group, adaptation aspect, FHH group, AO group, and SE group. The combination approach involved two rule types of assessment matrix (Figure 2). Assessment matrix 1 is to emphasize for identity the "real' occurrence, high vulnerability area. The combination rule is applied with climate risk simulation (CRS) between CP and RSL, the score of exposure aspect between CRS and CP, and the potential impact between exposure and sensitivity. Assessment matrix 2 is to emphasize the importance of adaptation, and it is used for the score of vulnerability, combined with potential impact and adaption. Finally, this article compares the vulnerability results with the types of agricultural land in law, to examine the relevant applicability and efficiency of different policies.

**Table 3.** Vulnerability Indicator System for Agricultural Land Use.

| Aspect | Group * | Item * | References |
|---|---|---|---|
| Expos-ure (E) | Disaster potential (DP) | Disaster potential distribution | Geographic Information Platform for National Spatial Planning of CPAMI |
| | Climate prediction (CP) | Rice impact analysis | Taiwan Climate Change Projection and Information Platform of NCDR |
| | Rising sea levels (RSL) | Risk of rising sea levels | Chen et al. (2001); EPA |
| Sensiti-vity (S) | Soil (SO) | Farmland production ability (FPA) | Taiwan Agricultural Land Information Service (TALIS) of Council of Agriculture (COA) |
| | Water (WA) | Distance to irrigation channels (DIC) Ratio of irrigation areas (RIA) | TALIS Agriculture, Forestry, Fishery, and Animal Husbandry Census |
| | Blocks (BL) | Farmland block areas (FBA) Scales of cultivated land (SCL) | |
| | Sensitive areas (SA) | Ecological conservation areas (ECA) | Geographic Information Platform for National Land Planning in CPAMI |
| Adapt-ation (A) | Farmer household characteristics (FHH) | Number of villager households/village area (NHH) | Agriculture, Forestry, Fishery, and Animal Husbandry Census |
| | | Age of person who commands cultivation (APC) | |
| | | Educational degree (ED) | |
| | | Average agricultural income of villager household (AAI) | |
| | Agricultural organization (AO) | Membership/number of people involved in agriculture (MPIA) | Annual statistics of farmers' association |
| | | Counts of production to sale unit (CPS) | |
| | | Number of people participating in practice courses/number of people involved in agriculture (NPP) | |
| | | Expenditure-to-revenue ratio of farmers' association (ERA) | |
| | | Rural regeneration community (RRC) | |
| | Socio-economic characteristics (SE) | Nurture ratio (NR) | Social Economic Database of the National Geographic Information System |
| | | Aging index (AI) | |

* Computing consequences and diagrams of all indicator groups and items are presented in Appendix A.

### 3.2.1. Exposure Criteria

In general, changes in climatic conditions and the frequency of occurrence for extreme climatic events have been used as exposure factors [7,8,10,11]. However, collecting relevant information including both the specified scale and details of climate impact, particularly at the local level, is

extremely difficult. Therefore, here, we mainly used the official report of farmland management issued by the agricultural association of the Taiwan government [6]. In the report, the exposure factors have been divided into three patterns: distribution of the current disaster potential, climate prediction, and rising sea levels.

Pattern 1, distribution of current disaster potential, describes real danger to territory, and evaluates the consequences of disaster potential distribution suggested by the Taiwan government. The consequences are concerned with areas located at sites with significant risks, such as specified water and soil conservation areas, security forest areas, river covered regions, coastal erosive areas, fault areas, and mudflow sensitivity areas. The consequences explain the various precarious positions and potential calamities of the present situation in Taiwan. As a result of the uncertainty of climate change, the possible risks of the present situation in these areas will continue to increase in number.

Pattern 2, climate prediction, focuses on the impact of growth cycles for crops and paddy rice by predicting climate scenarios A2, B1, and A1B in the IPCC AR4. The prediction data has been obtained from the project achievement of the Taiwan Climate Change Projection and Information Platform (TCCIP), with a data resolution of approximately $5 \times 5$ km, including two prediction periods (2020–2039 and 2080–2099). By combining various climatic contexts, the output for the possible impact of growth cycle extension for paddy rice was identified here.

Pattern 3, rising sea levels, examines the extension of impact under different scenarios, as well as the extension of impact as affected by several variables, including the degree of sea-level rise, the average elevation of the sea, the amplitude of the astronomical tide, and storm surge. The results have also been published in another technical document [16], with various scenarios of rising sea levels (increases of 2, 4, and 6 m). In this study, the scenario of a 2 m increase in sea level was applied for computing the visual extension of agricultural land.

To explore the effects of these patterns, this study adapted ranking rules to reclassify the impact levels into three equal parts. According to the reclassification, the degree of impact for each unit was defined as 1 for weak, 2 for medium, and 3 for strong. The reclassification was useful for distinguishing effects based on various factors in each analysis unit. These score modes and the classification method were applied throughout the analysis procedure in this study.

### 3.2.2. Sensitivity Criteria

The sensitivity aspect explores the sensitivity level of productivity for agricultural land, and includes four types of indicator groups: soil, water, blocks, and sensitive areas, as well as nine subcategories. Soil is the major focus for investigating the production ability of the agricultural land; however, different types of crops require different cultivation environments in terms of factors, such as climatic conditions and internal soil attributes. In this study, the major crop in Southern Taiwan, paddy rice, was used as the standard to represent the effects of the soil indicator, thereby avoiding excessive complexity.

Water resource management is also a crucial study area. Here, the correlation between farming and irrigation was considered in the water group. The blocks group mainly identified the properties and scales of the spatial unit of each agricultural land area. The sensitive area group depended on the geographical and environmental conditions based on the boundaries of the ecological conservation areas. By computing the mean of these criteria using GIS tools, the level of farmland vulnerability based on the sensitivity aspect could be measured.

### 3.2.3. Adaptation Criteria

The structure of the adaptation aspect was divided into three indicator groups: household characteristics of farmers, agricultural organizations, and socioeconomic characteristics. These three groups comprised the fragile properties of villager households, socioeconomic interactions between households, and social burdens in rural areas.

### 3.3. Data Source

Table 3 lists the types of items with the various scales, as well as references for the vulnerability indicator system used in this study. Regarding the exposure aspect, there are 3 items based on the open official data from central government organizations, such as National Spatial Planning, the Construction and Planning Agency, the Ministry of the Interior (CPAMI), the National Science and Technology Center for Disaster Reduction (NCDR), and the Environmental Protection Administration Executive Yun (EPA). The distribution of potential disaster areas is obtained from the second Taiwan National Land Use Survey. Rice impact analysis was applied the prediction of climatic condition from TCCIP, produced by NCDR. The risk of rising sea levels is based on the simulation results from the research project of EPA. The data for the sensitivity aspect was mostly based on the Taiwan Agricultural Land Information System, established by the Council of Agriculture (COA), Taiwan. Regarding the adaptation aspect, the data were obtained mainly from official statistics, such as the annual report of county statistics and the agricultural census.

## 4. Results: Vulnerability Assessment and Adaptation Strategies

Based on our analysis framework, assessment procedures were progressively oriented with the aspects of exposure, sensitivity, potential impact, adaptation, and final evaluation of the vulnerability of agricultural land. The results of each indicator are described as follows.

### 4.1. Exposure

The spatial distribution of the exposure indicator is shown in Figure 5 and Table 4. According to the results, medium exposure levels are mostly located in the central plain toward the western area of Taiwan, occupying more than 91% of the total agricultural land in Southern Taiwan. Areas of high exposure (only approximately 0.58% of the total area of the study area) are scattered among the hills and mountain regions located on the border between Yunlin and Chiayi, and also in southeast Tainan.

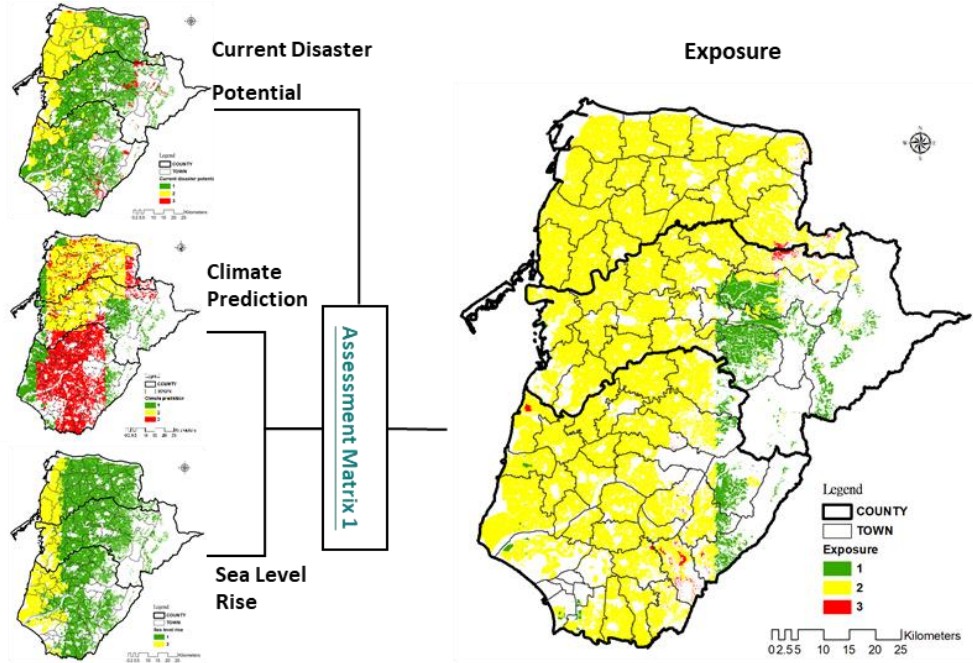

**Figure 5.** Spatial Distribution of the Exposure Indicator.

**Table 4.** Statistics of the Vulnerability Level Regarding the Exposure Aspect (Unit: ha).

| Vulnerability Levels for Exposure | Low | Medium | High |
|---|---|---|---|
| Yunlin | 21 | 97,430 | 313 |
| Percentage of the total area | (0.01%) | (31.53%) | (0.10%) |
| Chiayi | 18,097 | 65,584 | 615 |
| Percentage of total area | (5.86%) | (21.23%) | (0.20%) |
| Tainan | 6991 | 119,066 | 861 |
| Percentage of the total area | (2.26%) | (38.54%) | (0.28%) |
| Southern Taiwan | 25,109 | 282,081 | 1789 |
| Percentage of the total area | (8.13%) | (91.29%) | (0.58%) |

*4.2. Sensitivity*

Our results indicate that low sensitivity areas, most of which are located in the central plain of Yunlin and Chiayi, as well as in parts of northern Tainan, account for 65% of the total agricultural land (Figure 6; Table 5). Areas of high sensitivity (approximately 8%) are located near the southwest coast and eastern parts of the slope areas. Areas with medium sensitivity are scattered among the eastern hills and coastal areas in the western region. Areas of medium sensitivity account for approximately 27% of the total agricultural land.

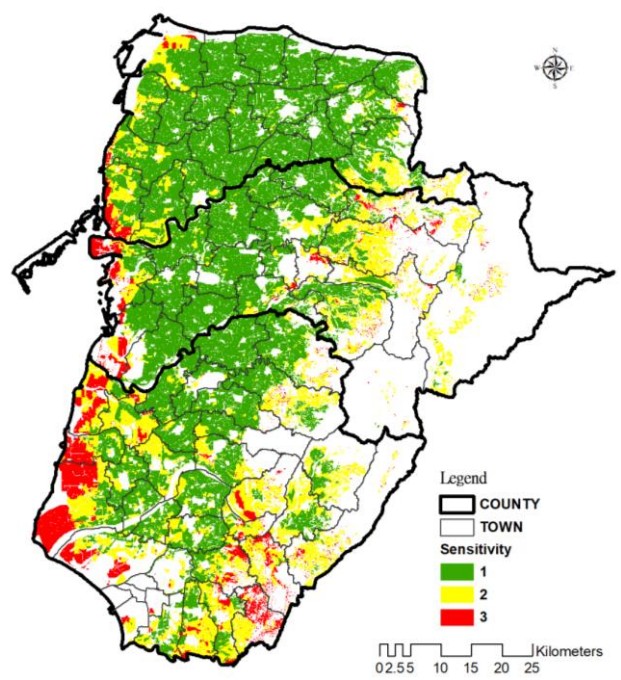

**Figure 6.** Spatial Distribution of the Sensitivity Indicator.

**Table 5.** Statistics of the Vulnerability Level regarding the Sensitivity Aspect (Unit: ha).

| Vulnerability Levels for Sensitivity | Low | Medium | High |
|---|---|---|---|
| Yunlin | 81,853 | 13,589 | 2322 |
| Percentage of total area | (26.49%) | (4.40%) | (0.75%) |
| Chiayi | 57,398 | 21,871 | 5027 |
| Percentage of total area | (18.58%) | (7.08%) | (1.63%) |
| Tainan | 62,101 | 46,946 | 17,871 |
| Percentage of total area | (20.10%) | (15.19%) | (5.78%) |
| Southern Taiwan | 201,353 | 82,407 | 25,221 |
| Percentage of total area | (65.17%) | (26.67%) | (8.16%) |

### 4.3. Potential Impact

Potential impact involves both the exposure and sensitivity aspects. Assessment matrix (AM)-I was used to integrate the two aspects (Table 6). AM-I emphasizes the effects of high exposure risk, and attempts to highlight low sensitivity vulnerability to recognize kernel production sites. By resembling the results of exposure vulnerability and sensitivity vulnerability, AM-I illustrates the distribution of agricultural land vulnerability regarding the potential impact aspect (Figure 7). Areas exhibiting high vulnerability to potential impact are mainly located in coastal areas, such as Yunlin, Tainan, and parts of areas near the hills in southeast Tainan. The results of overall spatial distribution were similar to the results of sensitivity vulnerability; however, the value of the central plains region shifted from medium to low vulnerability. The key areas with the low vulnerability of potential impact are located in eastern Chiayi, accounting for 2.04% of the entire agricultural land area (Table 7).

**Table 6.** Assessment matrix (AM)-I for Exposure and Sensitivity.

|  |  | Exposure | | |
| --- | --- | --- | --- | --- |
|  |  | **1** | **2** | **3** |
|  | 1 | 1 | 2 | 2 |
| **Sensitivity** | 2 | 2 | 2 | 3 |
|  | 3 | 2 | 3 | 3 |

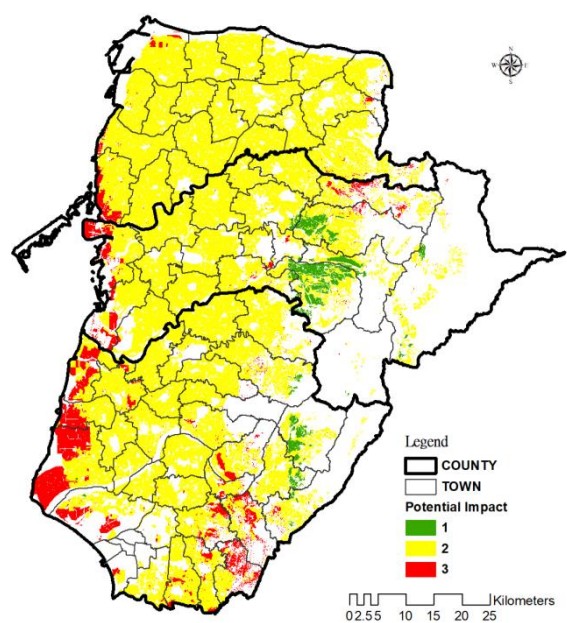

**Figure 7.** Spatial Distribution of the Potential Impact Indicator.

**Table 7.** Statistics of the Vulnerability Level regarding Potential Impact (Unit: ha).

| Vulnerability Levels for the Potential Impact | Low | Medium | High |
| --- | --- | --- | --- |
| Yunlin | 0 | 95,302 | 2462 |
| Percentage of the total area | (0.00%) | (30.84%) | (0.80%) |
| Chiayi | 6289 | 73,465 | 4542 |
| Percentage of the total area | (2.04%) | (23.78%) | (1.47%) |
| Tainan | 1928 | 107,276 | 17,714 |
| Percentage of total area | (0.62%) | (34.72%) | (5.73%) |
| Southern Taiwan | 8217 | 276,044 | 24,719 |
| Percentage of the total area | (2.66%) | (89.34%) | (8.00%) |

*4.4. Adaptation*

The results for adaptation demonstrate that regions with the high vulnerability of household characteristics of farmers are in coastal areas, and that the scale of the locations gradually decreases toward the eastern slope, indicating that the households of individual farmers residing in the plain and slope areas have stronger abilities to cope with climate change impact caused by certain characteristics. The agricultural organizations presented the overall development of local farming methods, according to their learning mechanisms for contacting their organizations. The results indicated that areas with low vulnerability are closer to the peri-urban areas. The socioeconomic characteristics of the analysis demonstrated that the dependency ratio is similar to the spatial structure of the aging index. The high vulnerability locations are centralized in boundaries between counties and aggregated in the southeastern rural area.

By integrating the criteria for the adaptation aspect, the spatial structure is presented in Figure 8. The high adaptation vulnerability areas are mainly located in coastal areas and on county borders. These regions cover 17 townships in Tainan, seven in Yunlin, and five in Chiayi. No significant differences were detected between the percentages for each level (Table 8); notably, the sites in central and eastern areas of Yunlin and Chiayi include complete blocks, with relatively low levels of adaptation vulnerability, whereas the edge townships of Tainan consistently demonstrate high adaptation vulnerability.

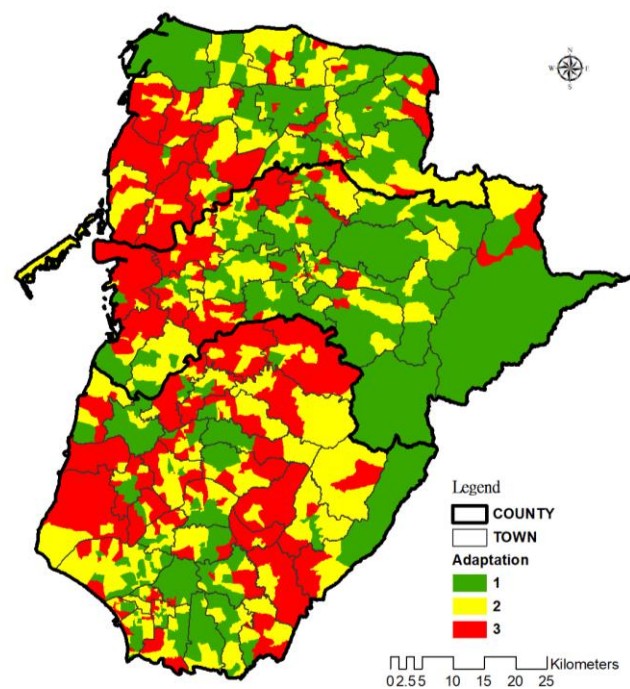

**Figure 8.** Spatial Distribution of the Adaptation Indicator.

**Table 8.** Statistics of the Vulnerability Level for the Adaptation Aspect (Unit: ha).

| Vulnerability Levels for Adaptation | Low | Medium | High |
|---|---|---|---|
| Yunlin | 27,221 | 38,230 | 32,312 |
| Percentage of the total area | (8.81%) | (12.37%) | (10.46%) |
| Chiayi | 36,198 | 27,192 | 20,906 |
| Percentage of the total area | (11.72%) | (8.80%) | (6.77%) |
| Tainan | 36,788 | 43,180 | 46,950 |
| Percentage of total area | (11.91%) | (13.98%) | (15.20%) |
| Southern Taiwan | 100,208 | 108,603 | 100,169 |
| Percentage of the total area | (32.43%) | (35.15%) | (32.42%) |

### 4.5. Analysis of Vulnerability Assessment for Farmland

AM-II was used for integrating the different criteria (Table 9). AM-II made the assumption that the potential impact was of equal importance to the adaptation aspect. According to the rule of AM-II, we further combined the results of potential impact and outcome-oriented by the adaptation aspect, by organizing the results shown in Figure 9. From the simulation results, areas with high levels of vulnerability occupy approximately 106,615 ha, accounting for 34.51% of the total agricultural land area. The locations are mostly located in western coastal areas, particularly in the two townships of Yunlin and seven of Chiayi, as well as some scattered areas in the central plain area. Additionally, areas with low levels of vulnerability occupy approximately 151,099 ha, of which 50,827ha are located in Tainan (highest frequency of areas with low vulnerability among all the territories; Table 10). In general, the location is closer to the central plains than to the eastern area. The results indicate that the central plain region to the east has a low level of vulnerability, implying an advantage for agricultural production, whereas the other regions comprise varied vulnerability levels. Concerning the climate change impact, the western coastal zone is a critical area. As a result of the distribution of the results, the demand for agricultural adaptation strategies could play a crucial and urgent role in Southern Taiwan.

**Table 9.** AM-II for Potential Impact and Adaptation.

|  |  | Potential Impact | | |
|---|---|---|---|---|
|  |  | **1** | **2** | **3** |
| **Adaptation** | 1 | 1 | 1 | 2 |
|  | 2 | 1 | 2 | 3 |
|  | 3 | 2 | 3 | 3 |

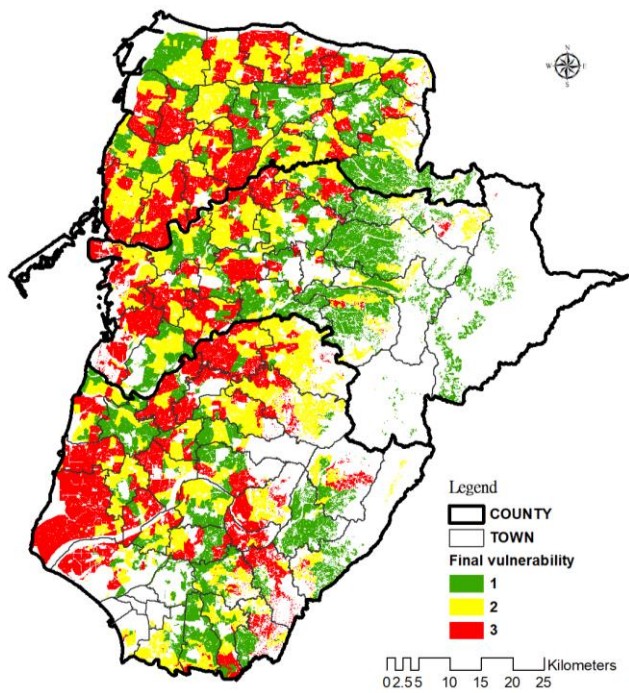

**Figure 9.** Spatial Distribution of Vulnerability.

**Table 10.** Statistics of the Different Levels of Vulnerability (Unit: ha).

| Levels of Vulnerability | Low | Medium | High |
|---|---|---|---|
| Yunlin | 26,666 | 37,612 | 33,485 |
| Percentage of the total area | (8.63%) | (12.17%) | (10.84%) |
| Chiayi | 36,534 | 25,460 | 22,302 |
| Percentage of the total area | (11.82%) | (8.24%) | (7.22%) |
| Tainan | 33,679 | 42,412 | 50,827 |
| Percentage of the total area | (10.90%) | (13.73%) | (16.45%) |
| Southern Taiwan | 96,880 | 105,485 | 106,615 |
| Percentage of total area | (31.35%) | (34.14%) | (34.51%) |

*4.6. Adaptive Strategies in Response to Climate Change*

The final vulnerability results identify the different characteristics of agricultural land in Southern Taiwan. This study attempted to establish some adaptive strategies for agricultural land for dealing with climate change by considering both the vulnerability of agricultural land and local agricultural policies as types of agricultural land in law. Each local agricultural policy and its related action was promoted in the specified location. Therefore, the pattern of adaptive strategies for agricultural land was recognized through the location-based relationship between vulnerability levels and local agricultural policies, such as farmland types in law. For the production ability of agricultural land, vulnerability, and properties of local agricultural policies, three types of adaptive strategies were proposed. The various types of spatial locations and pattern attributes are presented in Figure 10, Tables 11 and 12. The identifications of these types are described below.

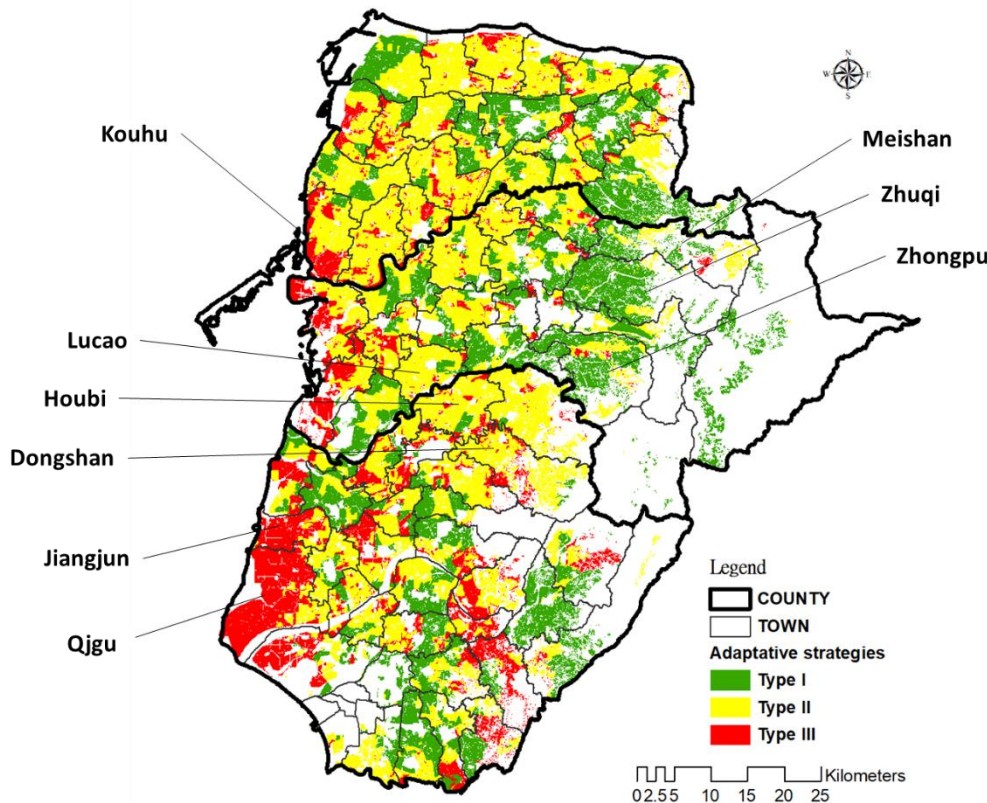

**Figure 10.** Spatial Distributions of the Adaptive Patterns of Agricultural Land.

**Table 11.** Proposed Adaptive Strategies for the Different Levels of Vulnerability.

| Type | Description | Long-Term Adaptive Policies | Short-Term Adaptive Actions |
|---|---|---|---|
| Type I: Enhancement of agricultural production | Areas mainly have low levels of vulnerability. Objectives: to maintain current production conditions and to strengthen production ability and efficiency. | ☑ To implement large basin management plans <br> ☑ To strengthen the efficiency of water resources <br> ☑ To develop and improve farming technology <br> ☑ To arrange workshops for cultivation technology | ☑ To arrange the cultivation period for paddy rice <br> ☑ To maintain the agricultural production environment <br> ☑ To promote pollution prevention actions <br> ☑ To improve irrigation facilities <br> ☑ To identify the agricultural management zone |
| Type II: Maintenance of agricultural production | Areas mainly have medium and high levels of vulnerability and more favorable production conditions, but also face the impact of climate change. Objective: To maintain the fundamental production function by considering food security. | ☑ To develop water-saving technology <br> ☑ To push forward large basin management plans <br> ☑ To strengthen the efficiency of water resource usage | ☑ To arrange the cultivation period for paddy rice <br> ☑ To improve irrigation facilities <br> ☑ To cultivate other heat-resistant crops |
| Type III: Conservation of agricultural environment | Areas mainly have medium and high levels of vulnerability and are mainly located in the coastal zones in the western areas and the hills in the southeastern areas. The production condition is relatively poor and easily affected by the climatic impact. Objective: mainly, to conserve the agricultural environment and its multifunctionality. | ☑ To continually monitor agricultural land <br> ☑ To complete ecological system planning and management of agricultural land <br> ☑ To implement agricultural land rest and the compensation mechanism | ☑ To adjust farming types <br> ☑ To maintain farmland rest and increase afforestation <br> ☑ To promote agricultural recreation <br> ☑ To protect rural heritage sites |

**Table 12.** Areas of Different Types with their Adaptive Strategies (Unit: ha).

| Suggested Adaptive Strategy Types | Type I | Type II | Type III |
|---|---|---|---|
| Yunlin | 26,666 | 59,493 | 11,604 |
| Percentage of the total area | (8.63%) | (19.25%) | (3.76%) |
| Chiayi | 36,534 | 38,645 | 9117 |
| Percentage of the total area | (11.82%) | (12.51%) | (2.95%) |
| Tainan | 33,679 | 57,975 | 35,264 |
| Percentage of the total area | (10.90%) | (18.76%) | (11.41%) |
| Southern Taiwan | 96,880 | 156,114 | 55,986 |
| Percentage of the total area | (31.35%) | (50.53%) | (18.12%) |

### 4.6.1. Type I: Enhancement of Agricultural Production

Type I farmland is mainly located in areas with low levels of vulnerability. As a result, it has only a slight impact, so climate change factors can be somewhat entirely disregarded in these areas. Excessively strict agricultural policies may produce interference for agriculture development. Thus, the concept of Type I farmland is to maintain the current production conditions and strengthen production ability and efficiency for farmland. This adaptive program is named the "Enhancement of agricultural production."

The overall objectives of the program are to maintain the original fertility of the farmland and strengthen its production function. To achieve these objectives, key adaptive strategies and actions were selected from all local agricultural policies. In short term, these areas focus on immediate actions, such as rearranging the rice cultivation period, maintaining the agricultural production environment, promoting pollution prevention actions, promoting irrigation facilities, and setting agricultural management zones.

In the long term, this program is concerned with the promotion and innovation of environmental techniques. These adaptive strategies have been adopted to push forward large basin management plans, strengthen the efficiency of water resource allocation, develop farming techniques, and present cultivation techniques workshops. The results indicate that Type I locations are commonly located in the central-to-east regions of Yunlin and Chiayi and the southern-to-east hill band of Tainan. Among these locations, the most complete blocks of farmland with the biggest areas are those in Chiayi (e.g., the Meishan, Zhuqi, and Zhongpu townships).

### 4.6.2. Type II: Maintenance of Agricultural Production

Type II farmland is mainly located in areas with medium and high levels of vulnerability. Considering food security, these farmlands should maintain their production functions. They encounter more intensive impact than do Type I farmland areas, so Type II farmland areas require the construction of an appropriately adaptive strategy to control their production abilities and quality in areas of qualified farmland and general farmland. The adaptive program is named the "Maintenance of agricultural production."

The key objectives are to promote adaptive capacity and safeguard land fertility. In these areas, short-term actions regarding the adjustment of crop production processes and techniques are emphasized. These actions include arranging the rice cultivation period, promoting irrigation facilities, and choosing heat-resistant crops. Long-term goals are concerned with the use of water resources. The relative adaptive strategies are to develop water-saving farming techniques, push forward large basin management plans, and strengthen the efficiency of water resource allocation.

This pattern has high connectivity, with an entire ratio of over 50% of farmland areas. These locations are mainly distributed on the central plains toward the west. This phenomenon is quite evident, particularly in areas of Chiayi and Tainan, such as the Lucao township, Houbi district, and Dongshan district. Type II farmland almost covers Yunlin in its entirety, accounting for 59,493 ha.

### 4.6.3. Type III: Conservation of Agricultural Environment

Type III farmland is mainly located in areas with medium and high levels of vulnerability. Type III farmland terrains can withstand quite an intense climate change impact. As a result of the relative weaknesses in the production condition, the cost of maintaining production function in these areas is relatively high. The adaptive program named the "Conservation of the agricultural environment" is concerned with the equilibrium of the multiple functions of farmland beyond production ability.

The objective of the program is to promote the adaptive capacity to conserve the multifunctional nature of the agricultural environment. The conservation of the multi-functionality of the agricultural environment provides the agricultural environment with nonproduction functions for water resource conservation, climate regulation, habitat conservation, and landscape management. In these areas, short-term actions involve the appropriate use of local assets, such as adjustment of farming types, farmland rest and afforestation, promotion of leisure agriculture, and protection of rural heritage.

The long-term goal is concerned with the overall planning of land use. The relative adaptive strategies are continual farmland monitoring, farmland ecological system planning and management, and the design of farmland rest and compensation mechanisms. As mentioned, Type III farmland areas are mainly located in western coastal and southeastern areas; however, Type III farmland areas are scattered around the entire study area. Most of these areas are surrounded by Type II areas, with those apparent being located in the Kuohu township in Yunlin and Jiangjun and Qigu districts in Tainan. In terms of area, the highest percentage share is Tainan City with 11.41%; this area is also the largest, accounting for 35,264 ha.

### 5. Discussion and Conclusions

From the perspective of the multifunctional nature of agricultural land, this study attempted to examine the characteristics of agriculture land by vulnerability assessment. The results of the vulnerability assessment have been used to explore the appropriate and efficient adaptive policies in Southern Taiwan. The contributions of this study are described as follows.

In terms of the vulnerability assessment framework, this study involved many approaches to moderate data and integrate computation results. For instance, the assessment matrices were employed to establish rules for combining consequences from different sectors. These procedures were limited by the data format, uncertainties in the relationships between criteria, and a lack of information; however, despite these limitations, through the conceptual integration mechanism, the matrices could still effectively simulate the relative locations of vulnerable agricultural land, and subsequently develop follow-up strategies.

According to the results of the vulnerability assessment of agricultural land, the entire study region can be divided into three patterns: Pattern 1, located in the western coastal zone, filled with various attributes of high vulnerability; Pattern 2, distributed on the central plain region in the east, particularly in Yunlin and Chiayi, with most of these regions having complete blocks of agricultural land with low levels of vulnerability; and Pattern 3, located in the central plain region to the west, a region in which areas with various vulnerability levels are scattered throughout. This distribution is affected by the results of the adaptation aspect. Thus, the spatial structure of vulnerability in the agricultural land in Southern Taiwan ranges from the southwest coast with high vulnerability, to the northeast area with low vulnerability.

This study constructed adaptive strategies for agricultural land through a linkage between agricultural land vulnerability and agricultural production ability. By producing a matrix for agricultural land, three types of adaptive strategies were proposed. The objectives of Type I farmland are to maintain the original land fertility and strengthen its production function; this farmland is distributed throughout the northeast plain region. The key objectives of Type II farmland are to promote adaptive capacity and safeguard land fertility; this farmland is distributed throughout the central plain region in the west, with various levels of agricultural land vulnerability. The objective of Type III farmland is to promote the adaptive capacity to conserve the multifunction nature of the

agricultural environment, emphasizing the conservation of the entire environment; the major location of this farmland is the southwestern coastal area.

Finally, agricultural land is one of the essential elements of agriculture. Emphasizing the multiple values of the nature of agricultural land will help to enhance the possibility and application of policy inputs. The main purpose of the application of vulnerability assessment is not to explicitly describe the status of agricultural land to climate change, but to help local government and farmers to identify the critical area and to discussing the appropriated adaptive policies and agriculture future in the APFs process. The consequence of vulnerability is a relative score, and it simplifies the complex values for local people. Local people can feel a real difference between individuals and neighbors. That creates the possibility to connect local government and local farmer, and to generate consequences. In addition, by the characteristics of agriculture land location, adaptive policies would have new opportunities to integrate, to efficiently benefit, and to create a new development direction.

**Author Contributions:** Conceptualization, S.-L.C. and C.-J.L.; methodology, K.-C.H. and S.-L.C.; software, K.-C.H.; validation, S.-L.C. and K.-C.H.; data curation, K.-C.H. and C.-H.T.; writing—original draft preparation, K.-C.H.; writing—review and editing, S.-L.C.; visualization, K.-C.H.; supervision, S.-L.C.; project administration, S.-L.C. and C.-J.L. All authors have read and agreed to the published version of the manuscript.

**Funding:** This research was funded by the Council of Agriculture, Taiwan (R.O.C) under grant number 101-AgriTech-14-1-1-Plan-Q2. The APC was funded by the Council of Agriculture, Taiwan (R.O.C) under grant number 109-AgriMana-1.3-Plan-02.

**Acknowledgments:** This paper is a partial result of the research project 101-AgriTech-14-1-1-Plan- Q2. Financial support for this research was granted by the Council of Agriculture, Taiwan (Republic of China), which is gratefully acknowledged by the authors.

**Conflicts of Interest:** The authors declare no conflict of interest.

## Appendix A

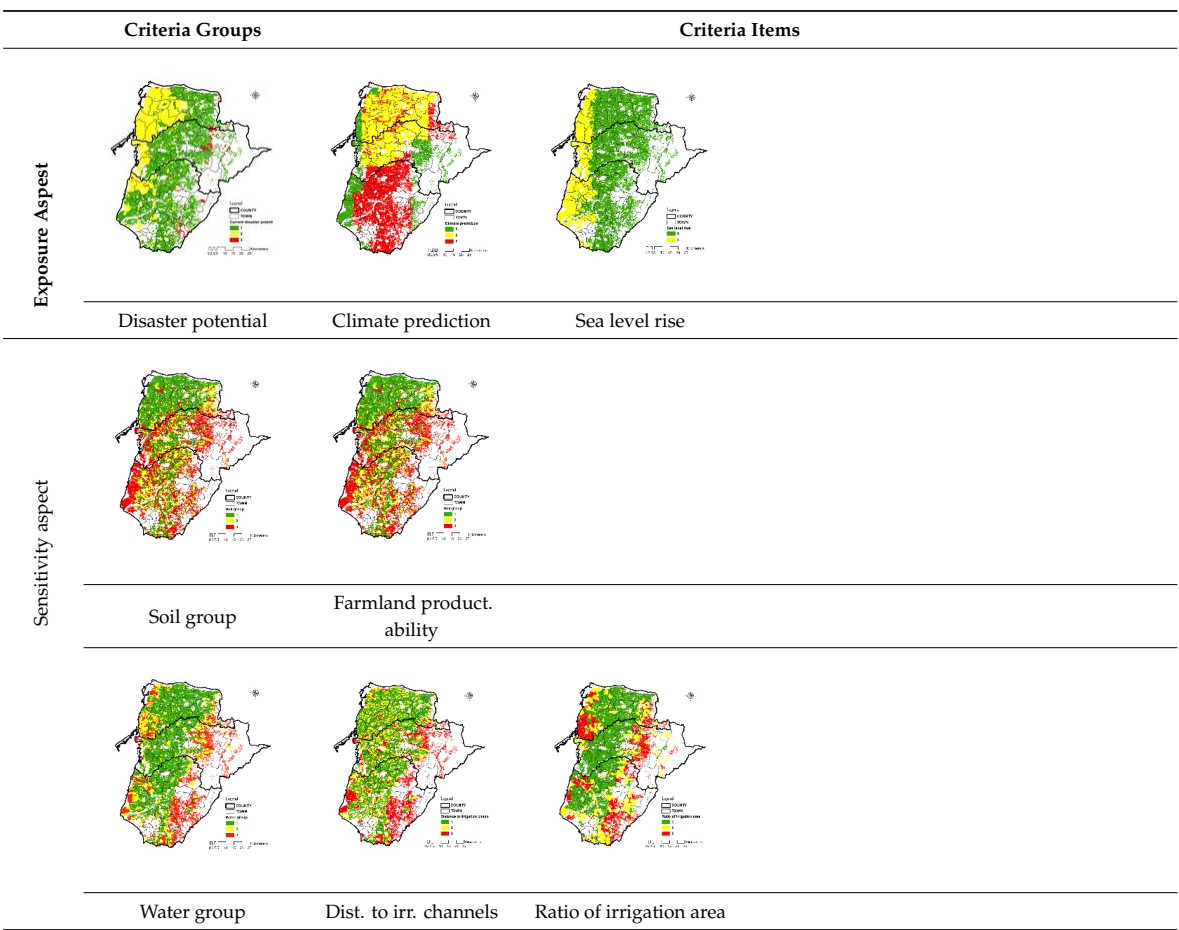

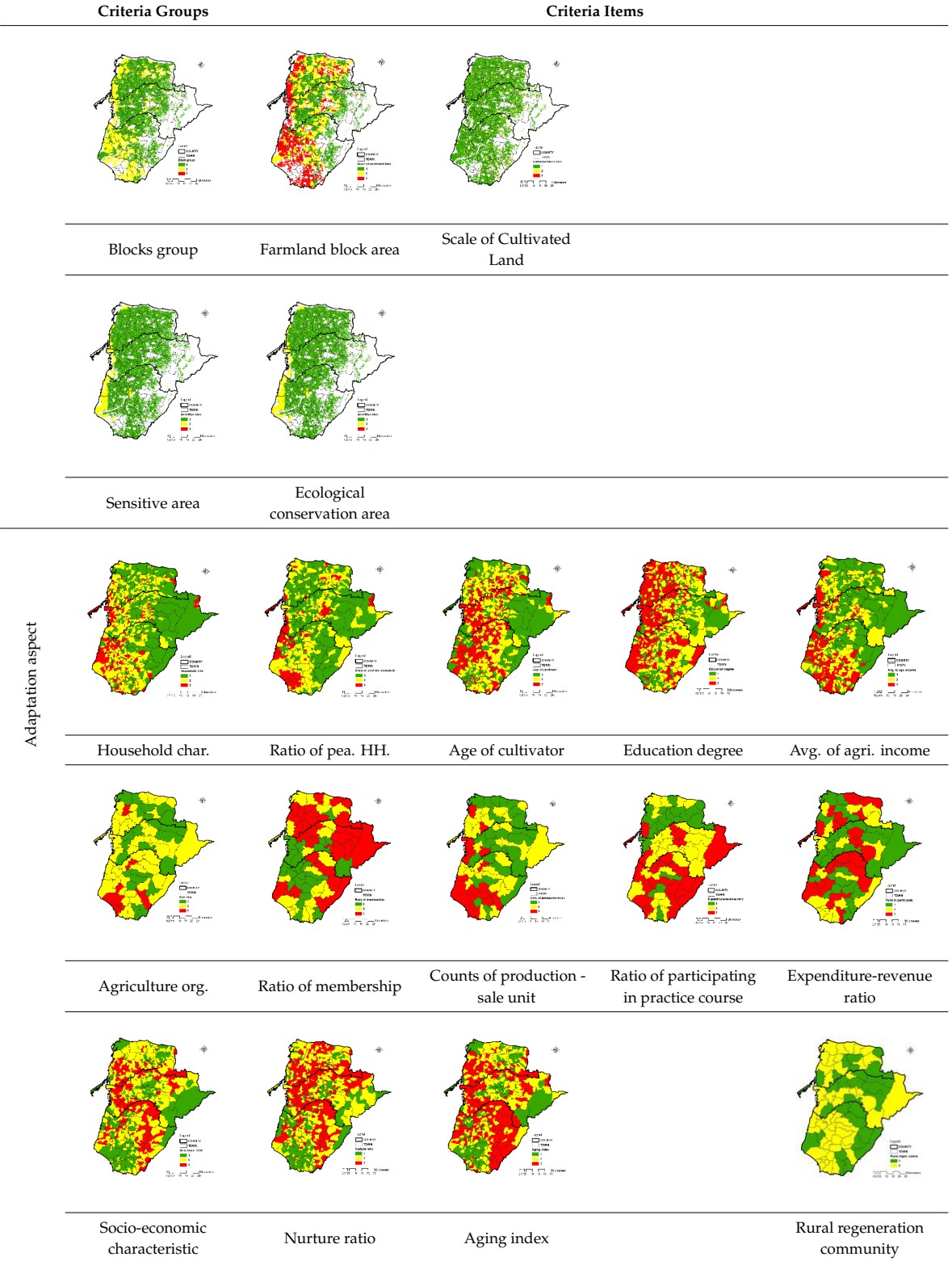

| Criteria Groups | | Criteria Items | | |
|---|---|---|---|---|
| | Blocks group | Farmland block area | Scale of Cultivated Land | |
| | Sensitive area | Ecological conservation area | | |
| Adaptation aspect | Household char. | Ratio of pea. HH. | Age of cultivator | Education degree | Avg. of agri. income |
| | Agriculture org. | Ratio of membership | Counts of production - sale unit | Ratio of participating in practice course | Expenditure-revenue ratio |
| | Socio-economic characteristic | Nurture ratio | Aging index | | Rural regeneration community |

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
