# Peer review of "Vulnerability Assessment and Adaptation Strategies for the Impact of Climate Change on Agricultural Land in Southern Taiwan"

_sustainability, doi:10.3390/su12114637_

Round 1

Reviewer 1 Report

This is a very interesting paper that is well written and presents a good research design. The authors map land areas (agricultural lands primarily) in Taiwan and rate the lands for climate change risk. Adaptation strategies are offered for high areas. I attach a draft of the MS with track changes and a few comments. The two areas I think they can make more clear are: water resource impacts and weighting among criteria. There are several places where water resources are mentioned, but there is no mention of erosion control, riparian area management, downstream movement of nutrients or pesticides or other impacts. I have offered comments in places they might consider mentioning something. Second, they are using a risk index that includes variables like climate sensitivity and adaptation. The way they present the material, it appears that their risk factor is the product of four, equally weighted variables. Many authors have done similar work, and the variables can be weighted differently. There are good arguments for and against weighting. I suggest that these authors just state explicitly that the four variables are equally weighted.

Reviewer 2 Report

This study attempted to explore the relationships between vulnerability assessment and adaptive strategy through the operation of vulnerability assessment and construction of adaptive strategies in Southern Taiwan. In the terms of the vulnerability assessment framework, this study involved many approaches to moderate data and integrate computation results. Regarding the assessment of agricultural land study proposed the division of region into three patterns.

The structure of the manuscript is is good and produces results that I had hard time to confirm in the black and white figures. The use of English at many points is poor and needs more work (see my comments in the attached manuscript). This is the reason I recommend major revision before the acceptance of this study.

Reviewer 3 Report

Interesting paper about the relationship between vulnerability assessment and adaptation strategies for the potential impact of climate change. The paper is relevant to the aims and scope of the journal.

I have some issues with the methodology and related cartography.

  1. It is not clear which ponderation is attributed to each variable when they are overlapped. As an example when you calculated the exposure indicator the weigh attributed to Climate Prediction and Sea Level Rise is the same?  Or when the potential impact indicator is calculated the same weigh is attributed to sensibility and exposure? Please clarify the methodology used concerning the ponderation/ weigh ascribed to each variable;
  2. Most of maps legend has no read and needs to be revised since sometimes the dark scale corresponds to the level 3 and in other maps correspond to the level 2. There are several maps with white areas and just one map is in greyscale. Which means the white color: no data?

The discussion/conclusion is very poor. Improve the discussion/interpretation of the results considering similar studies. In the conclusion highlight, for instance, the novelty of the approach benefits provided, and open research perspectives and the study’s limitations/caveats.

Round 2

Reviewer 3 Report

Just a comment: the figures that have only two classes must use the corresponding colour, i.e., green (1) and yellow (2, instead of red colour that corresponding to the  third class)
